# Effect of Dietary Vitamin C Supplementation on Growth Performance and Biochemical Parameters in Grower Walleye Pollock, *Gadus chalcogrammus*

**DOI:** 10.3390/ani14071026

**Published:** 2024-03-28

**Authors:** Ki Wook Lee, Hae Kyun Yoo, So-Sun Kim, Gyeong Sik Han, Min Min Jung, Hee Sung Kim

**Affiliations:** 1Aquaculture Industry Research Division, East Sea Fisheries Research Institute, National Research Institute of Fisheries Science, Gangneung 25435, Republic of Korea; 1eekw@korea.kr (K.W.L.); sealeader@korea.kr (H.K.Y.); ssokim81@korea.kr (S.-S.K.); gshan0223@korea.kr (G.S.H.); jminmin@korea.kr (M.M.J.); 2Department of Marine Biology and Aquaculture, Gyeongsang National University, Tongyeong 53064, Republic of Korea

**Keywords:** walleye pollock, vitamin C, growth performance, biochemical parameters

## Abstract

**Simple Summary:**

Our study showed that a vitamin C (VC) supplementation in grower walleye pollock diet improved growth performance and plasma superoxide dismutase activity. Also, dietary VC significantly enhanced feed utilization, body protein content, and plasma growth hormone levels. Moreover, broken-line regression analysis considering weight gain revealed that the optimal dietary VC level for grower walleye pollock was 156.42 mg kg^−1^ diet.

**Abstract:**

The optimal dietary vitamin C (VC) levels for walleye pollock (*Gadus chalcogrammus*) remain undefined. This study aimed to assess the effect of dietary VC levels on the growth performance and biochemical parameters of grower walleye pollock and determine the optimal VC level for their diet. Six experimental diets (VC0, VC1, VC3, VC5, VC7, and VC10) with VC levels of 3.24, 21.92, 63.31, 101.42, 145.46, and 202.51 mg kg^−1^ diet, respectively, were fed to fish (initial mean weight: 173.5 ± 0.31 g) for 8 weeks. At the end of the feeding trial, fish fed the VC7 and VC10 diets exhibited significantly higher growth (final body weight, weight gain, and specific growth rate) and improved feed utilization (feed efficiency and protein efficiency ratio) compared with fish fed the VC0 diet (*p* < 0.05). The VC3–VC10 diets significantly reduced plasma superoxide dismutase (SOD) activity (*p* < 0.05). Compared with the VC0 group, fish fed the VC7 and VC10 diets showed significantly elevated growth hormone and insulin-like growth factor-1 levels in plasma (*p* < 0.05). In conclusion, dietary VC supplementation in walleye pollock improved growth performance and SOD activity. Moreover, broken-line analysis on weight gain indicated that the optimal dietary VC level for grower walleye pollock was approximately 156.42 mg kg^−1^ diet.

## 1. Introduction

Walleye pollock (*Gadus chalcogrammus*) is widely distributed in the North Pacific Ocean, with a range extending from the northern coast of Japan and the Okhotsk Sea to the Bering Sea and the Gulf of Alaska [1,2]. Walleye pollock is a globally important commercial fish species, known for its mild-flavored white meat and versatility in various food products [3]. It is commonly used in processed fish products, including surimi, fish sticks, and fillets [4]. Walleye pollock is a widely consumed fish in Korea, popular for its mild flavor and versatility in various culinary preparations [5]. In response to declining catches, Korea is actively advancing reliable aquaculture production technology [6,7] to promote conservation efforts and sustainably manage this species.

A key strategy for ensuring stable fish culture production involves the development of nutritionally balanced feed [8]. Therefore, identifying the nutritional requirements of walleye pollock and formulating a highly efficient diet to enhance their survival, growth, and general health is crucial. Although optimal dietary protein and lipid levels for walleye pollock have been established at 50%–55% [9] and 8%–17% [10], respectively, information on dietary ascorbic acid [vitamin C (VC)] levels remains absent for this species.

VC, a water-soluble micronutrient, plays diverse roles in fish physiology [11]. It contributes to collagen biosynthesis [12], carnitine and norepinephrine production essential for growth [13], enhanced nutrient absorption [14,15], immune system support [16,17], antioxidant properties [18,19], and stress reduction [20]. Teleost fish species lack the capacity to synthesize their own VC, relying on obtaining it from their diet [21]. Thus, ensuring adequate VC levels in fish feed is imperative for providing essential nutrients and optimizing health in these species. However, variations arise due to factors such as fish species, developmental stage, fish size, VC type, feed formulation, feeding behavior, and the fish’s culture environment [13,21]. Hence, it is imperative to conduct fish-specific research to determine the optimum VC level in feeds. Prior investigations have established the optimal VC levels for the following species: grass carp (*Ctenopharyngodon idella*) at 92.8 to 129.8 mg kg^−1^ [22], cobia (*Rachycentron canadum*) at 13.6 mg kg^−1^ [23], large yellow croaker (*Pseudosciaena crocea*) at 28.2 mg kg^−1^ [24], juvenile Chu’s croaker (*Nibea coibor*) at 71.5 to 150.3 mg kg^−1^ [18]. Therefore, this study aimed to determine the optimal dietary VC levels for grower walleye pollock and investigate the effects of various VC levels on the growth, feed utilization, body composition, and biochemical indices of this species.

## 2. Materials and Methods

### 2.1. Experimental Diets

Six experimental diets were formulated, VC0 (control), VC1, VC3, VC5, VC7, and VC10, containing increasing VC levels: 0%, 0.01%, 0.03%, 0.05%, 0.07%, and 0.10%, respectively (Table 1). The VC used in these diets was supplied in the form of L-ascorbyl-2-monophosphate (Sigma-Aldrich, Darmstadt, Germany) owing to its superior heat resistance compared with unprotected VC and its high bioavailability to fish [25]. Table 1 shows the formulation and proximate composition of the experimental diets. Actual VC levels were 3.24, 21.92, 63.31, 101.42, 145.46, and 202.51 mg kg^−1^ in VC0, VC1, VC3, VC5, VC7, and VC10, respectively. Jack mackerel meal, krill meal, and fermented soybean meal were used as protein sources, whereas fish oil served as the lipid source. All dry ingredients were thoroughly blended using a commercial kitchen mixer, the fish oil was gradually incorporated while continuously mixing, and water was added to form a moist dough. A laboratory pellet extruder (SL Machinery, Incheon, Republic of Korea) was used to pelletize the ingredients, followed by oven drying at 30 °C. The experimental diets were then stored at −20 °C until use.

### 2.2. Preparation of Experimental Fish and Conditions

Prepared-for-use grower walleye pollock were sourced from the East Sea Fisheries Research Institute of the National Institute of Fisheries Science (Gangneung-si, Republic of Korea), where the fish were artificially hatched and raised to the desired initial size prior to the experiment. In total, 468 walleye pollock (average weight: 173.5 g) were randomly stocked into 12,400 L flow-through circular tanks (water volume: 300 L; 26 fish per tank) in triplicate. Tanks received sand-filtered seawater at a rate of 5.4 L min^−1^. The fish were hand-fed the experimental diets twice daily (09:00 and 17:00) to apparent satiation for 56 days. Daily monitoring of survival rate was conducted, and dead fish were removed and weighed immediately. Water temperature, dissolved oxygen, and salinity were measured daily, with average values of 7.5 ± 0.6 °C, 8.5 ± 1.2 mg/L, and 30.1 ± 2.5 psu, respectively. Light intensity on the water surface during the feeding trial was recorded as 20 lux. A 12:12 h light:dark photoperiod was maintained.

### 2.3. Sample Collection

In each tank, a fish count and total weight measurement was taken prior to calculating final body weight, survival rate, specific growth rate (SGR), feed consumption, feed efficiency (FE), and protein efficiency ratio (PER) at the end of the 56-day feeding trial following a 1-day starvation period. To determine final body proximate composition, five fish from each tank were randomly collected and frozen. Additionally, three fish from each tank were randomly chosen for blood sampling from the caudal vein using a heparinized syringe. Plasma was collected as separate aliquots following centrifugation (7500 rpm for 10 min) and stored at −80 °C for subsequent antioxidant capacity and growth-related hormone analyses.

### 2.4. Proximate Composition Analysis

Proximate compositions of the experimental diets and pooled whole-bodies were analyzed following the Association of Official Agricultural Chemists method [26]. Crude protein content was determined using the Kjeldahl method, employing automated Kjeldahl equipment (Buchi, Flawil, Switzerland). Crude lipids were measured using a Soxhlet extractor (VELP Scientifica, Milano, Italy). Moisture content was assessed following oven drying at 105 °C for 24 h, whereas ash content was measured using a muffle furnace at 600 °C for 4 h. VC concentration in the diets was determined using high-performance liquid chromatography (Agilent 1200 Series HPLC; Agilent Technologies, Anaheim, CA, USA).

### 2.5. Biochemical Analysis

An automated blood analyzer was used to evaluate plasma levels of aspartate aminotransferase (AST), alanine aminotransferase (ALT), total protein (TP), glucose (GLU), and total cholesterol (TCHO) (FUJI DRI-CHEM NX400i, Minato, Japan). Superoxide dismutase (SOD) activity in plasma was determined using commercially available assay kits (Cayman Chemical Company, Ann Arbor, MI, USA) following the manufacturer’s instructions. Plasma growth hormone (GH) and insulin-like growth factor (IGF-1) concentrations were measured using commercial fish ELISA kits (MyBiosource Co., San Diego, CA, USA) according to the manufacturer’s instructions. All samples were taken in triplicate.

### 2.6. Calculation and Statistical Analysis

Growth performance and feed utilization were calculated as follows: Survival (%) = (number of fish at the end of the trial/number of fish at the beginning of the trial) × 100;
Weight gain (WG; g/fish) = final body weight − initial body weight;
SGR (%/day) = [(ln final weight of fish − ln initial weight of fish)/days of trial] × 100;
Feed intake (FI, g/fish) = total dry feed intake/fish;
FE = weight gain of fish/feed consumed;
PER = weight gain of fish/protein consumed.

Before statistical analysis, the Kolmogorov–Smirnov test and Levene’s test were used to examine the normality of distribution and homogeneity of variances between distinct treatments, respectively. Percentage values were arcsine-transformed before analysis. One-way ANOVA and Tukey’s HSD test were used to analyze the significance of differences among treatment means. The optimal VC concentration was estimated using broken-line regression analysis. Additionally, correlations between WG and GH concentrations were determined using multiple regression analysis. All analyses were performed using SPSS version 27.0 software (SPSS Inc., Chicago, IL, USA).

## 3. Results

### 3.1. Growth and Feed Utilization 

Table 2 presents the growth and feed utilization parameters of grower pollock fed experimental diets with different VC levels for 8 weeks. Survival rates were 97.9%–100.0%, exhibiting no differences among treatments (*p* > 0.05). The final weight (g/fish) of fish fed VC7 was significantly higher than that of fish fed VC0 and VC1 (*p* < 0.05) but did not significantly differ from that of fish fed VC3, VC5, and VC10 (*p* > 0.05). WG in the VC7 and VC10 groups was significantly higher than that in the VC0 and VC1 groups (*p* < 0.05) but did not differ from that in the VC3 and VC5 groups (*p* > 0.05). The SGR of fish fed VC1–VC10 was significantly higher than that of fish fed VC0 (*p* < 0.05). The FI of fish fed VC3, VC7, and VC10 was significantly higher than that of fish fed VC0 (*p* < 0.05). The PER in the VC7 and VC10 groups was significantly higher than in the VC0–VC3 groups (*p* < 0.05). However, FE was not significantly affected by dietary VC supplementation (*p* > 0.05). Broken-line models based on WG revealed the optimal dietary VC supplementation level as 156.42 mg kg^−1^ (Figure 1).

### 3.2. Proximate Composition and Hematological Parameters

Table 3 presents whole-body proximate compositions. Whole-body crude protein content in fish fed VC7 and VC10 was significantly higher than that in fish fed VC0–VC3 (*p* < 0.05) but did not differ significantly from that in fish fed VC5. Moisture, crude protein, crude lipid, and ash content in pollock whole-bodies was 74.1–75.4%,16.3–18.0%, 6.3–6.7%, and 2.1–2.3%, respectively. Dietary VC did not affect moisture, crude lipid, and ash content in pollock whole-bodies.

Table 4 shows the plasma chemistry of pollock. AST, ALT, TCHO, GLU, and TP levels were 18.6–43.8 U/L, 6.1–7.7 U/L, 280.0–337.9 mg/dL, 79.1–93.9 mg/dL, and 3.9–4.4 g/dL, respectively. Hematological parameters in plasma were not significantly affected by dietary VC levels (*p* > 0.05).

### 3.3. SOD Activity and Growth-Related Hormone Parameter

Figure 2 shows SOD activity in the plasma of pollock. Plasma SOD activity in pollock fed VC5–VC10 was significantly lower than that in pollock fed VC0 and VC1 (*p* < 0.05) but did not differ significantly from that in pollock fed VC3 (*p* > 0.05). Figure 3 and Figure 4 present GH and IGF-1 concentrations in the plasma of pollock, respectively. Plasma GH concentration in pollock fed VC7 and VC10 was significantly higher than that in pollock fed VC0–VC5 (*p* < 0.05). Plasma IGF-1 concentration in fish fed VC7 and VC10 was significantly higher than that in fish fed VC0–VC3 (*p* < 0.05) but did not differ significantly from that in fish fed VC5 (*p* > 0.05). Furthermore, significantly positive linear correlations between walleye pollock WG and growth-related hormone (GH and IGF-1) levels are shown in Figure 5 and Figure 6 (*p* < 0.05).

## 4. Discussion

Numerous studies have highlighted the growth-promoting effects of dietary VC in various fish species [15,18,27,28]; however, the vitamin nutrition of walleye pollock had not been investigated previously. This study evaluated the growth and feed utilization of grower pollock fed with a VC-supplemented diet, comparing treatment groups with the control (VC0) group. Results revealed that fish fed VC7 (145.46 mg kg^−1^) and VC10 (202.51 mg kg^−1^) exhibited significantly higher final weight, WG, SGR, FI, FE, and PER compared with those fed VC0 (3.24 mg kg^−1^). These results indicate that dietary VC supplementation improved FI, FE, and PER, thereby enhancing final weight, WG, and SGR. Similar observations have been reported in other fish species, including large yellow croaker [24], largemouth bass (*Micropterus salmoides*) [13], mahseer fish (*Tor putitora*) [28], Chu’s croaker [18], Nile tilapia (*Oreochromis niloticus*) [29], and coral trout (*Plectropomus leopardus*) [19]. Dawood and Koshio [21] found that VC inclusion in the diet of aquatic animals enhances FI and feed utilization, leading to improved growth performance. Additionally, VC serves as an essential cofactor in the proline and lysine hydroxylation process during collagen metabolism, playing a vital role in bone and skin production and contributing to overall growth [12].

Based on walleye pollock WG, the optimal dietary VC level was 156.42 mg kg^−1^ diet. This level aligns with observations in other fish species, such as grass carp (129.8 mg kg^−1^) [22], yellow catfish (*Pelteobagrus fulvidraco* Richardson: 114.5 mg kg^−1^) [30], and yellow drum (*Nibea albiflora*: 142.2 mg kg^−1^) [31]. However, the observed level was higher than that reported for golden pompano (*Trachinotus ovatus*: 49.73 mg kg^−1^) [32], largemouth bass (102.6 mg kg^−1^) [13], large yellow croaker (28.2 mg kg^−1^) [24], cobia (13.6 mg kg^−1^) [23], and striped catfish (*Pangasianodon hypophthalmus*: 46 mg kg^−1^) [33] but lower than that reported for Wuchang bream (*Megalobrama amblycephala* Yih: 700 mg kg^−1^) [34], loach (*Misgurnus anguillicaudatus* Cantor: 226.2 mg kg^−1^) [35], and Nile tilapia (400 mg kg^−1^) [29]. The variation in optimal dietary VC levels across fish species can be attributed to metabolic activity [36], feeding habits [37], nutrient composition of feed, fish size [13], and the type of VC used [25,38].

VC plays a crucial role in protein metabolism and animal health [39,40,41]. In this study, whole-body crude protein content in fish fed VC7 (145.46 mg kg^−1^) and VC10 (202.51 mg kg^−1^) exceeded that in fish fed VC0 (3.24 mg kg^−1^). Biswas et al. [42] found that VC, acting as a cofactor in proline and lysine hydroxylation, enhances protein synthesis, protein deposition, and growth performance. Consequently, higher protein levels in the whole-body of walleye pollock contributed to muscle development and energy provision, resulting in improved growth performance. To enhance the growth and feed utilization of grower walleye pollock, it is recommended that the diet be supplemented with VC at 145.46 and 202.51 mg kg^−1^, respectively.

Hematological parameters serve as dependable indicators for evaluating the health and nutritional condition of aquatic organisms [21]. In this study, the health and nutritional condition of walleye pollock remained normal across all dietary treatment groups, evidenced by nonsignificant changes in AST, ALT, TCHO, GLU, and TP levels.

SOD plays a crucial role in eliminating free radicals, acting as a vital defense mechanism against oxidative stress [43]. SOD facilitates the transformation of the superoxide anion into hydrogen peroxide, which is subsequently catalyzed [44]. Several studies have demonstrated that VC supplementation impacts antioxidant enzyme activity, reducing oxidative damage caused by reactive oxygen species (ROS) [30,44,45]. Harsij et al. [46] reported a reduction in SOD activity in rainbow trout (*Oncorhynchus mykiss*) with VC supplementation, mitigating ROS-induced oxidative damage. Additionally, the mRNA expression levels of SOD in loach were markedly reduced through VC administration, leading to a decrease in SOD activity [35]. Similarly, the present study revealed a notable reduction in plasma SOD activity in walleye pollock due to VC supplementation, highlighting VC’s efficacy in eliminating ROS.

Growth hormone levels are directly associated with the growth of bones, muscles, and other body tissues. They indirectly contribute to animal evolution by stimulating the production and secretion of somatomedins, such as IGF-1 [47]. Multiple organs secrete IGF-1, a mitogenic hormone that regulates body growth in vertebrates and is influenced by nutritional conditions [48]. Duran et al. [49] noted that VC supplementation accelerates and advances myogenesis in fish myoblasts and influences IGF-1, a key factor in muscle growth, promoting downstream cascades that culminate in the activation of the mechanistic target of rapamycin and other processes integrating signals from nutrients, energy status, and growth factors, among other functions. As shown in previous studies, various fish species have displayed similar trends. For instance, an investigation into juvenile largemouth bass revealed the upregulation of IGF-1 and GH mRNA expression with elevated dietary VC levels [15]. The findings of the present study align with the aforementioned research, notably in the VC7 and VC10 treatment groups, where plasma GH and IGF-1 concentrations exhibited consistent trends. Additionally, the current study established a significant positive correlation between GH and IGF-1 concentrations and WG in walleye pollock administered diets supplemented with various VC levels. Elevated GH and/or IGF-1 levels have been consistently correlated with enhanced growth performance in various fish species [28,50,51,52]. These observed effects may be attributed to the influence of VC on elevating GH levels in the blood, enhancing intestinal morphology, and improving intestinal absorptive capacity in fish [53].

## 5. Conclusions

In conclusion, this study indicated that dietary VC significantly enhances feed utilization, body protein content, and GH levels, promoting growth and improving plasma SOD activity in walleye pollock. Broken-line regression analysis considering WG revealed that the optimal dietary VC level for grower walleye pollock was 156.42 mg kg^−1^ diet. These findings provide clear guidance for incorporating VC into the diet of walleye pollock to enhance their culture production.

## Figures and Tables

**Figure 1 animals-14-01026-f001:**
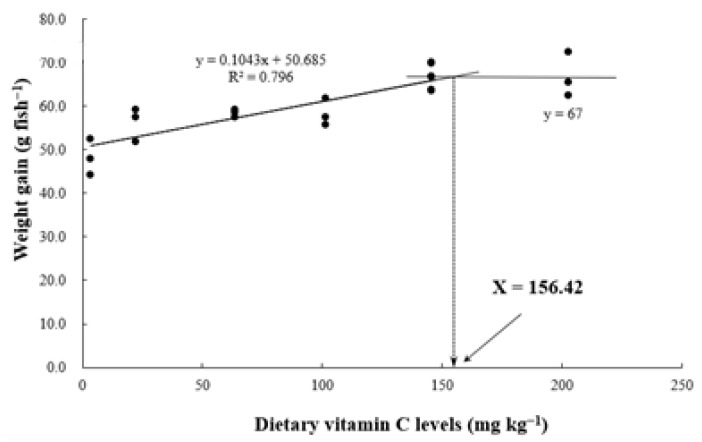
Broken-line analysis of the relationship between dietary vitamin C and weight gain of grower walleye pollock.

**Figure 2 animals-14-01026-f002:**
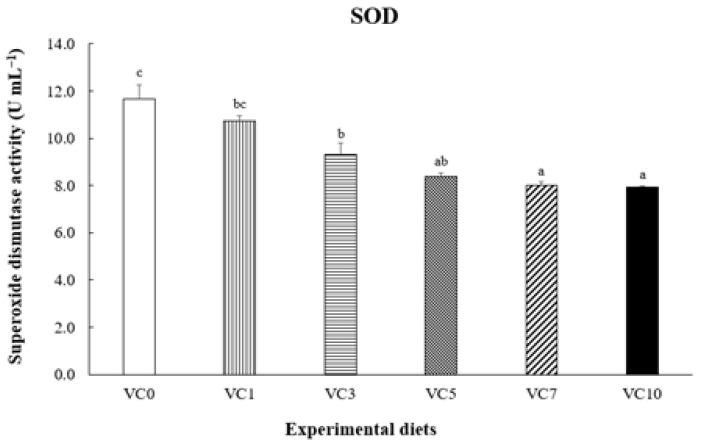
Dietary effect of supplementation of vitamin C on the superoxide dismutase (SOD) activity in plasma of grower walleye pollock. Different letters on column indicated significant differences (*p* < 0.05).

**Figure 3 animals-14-01026-f003:**
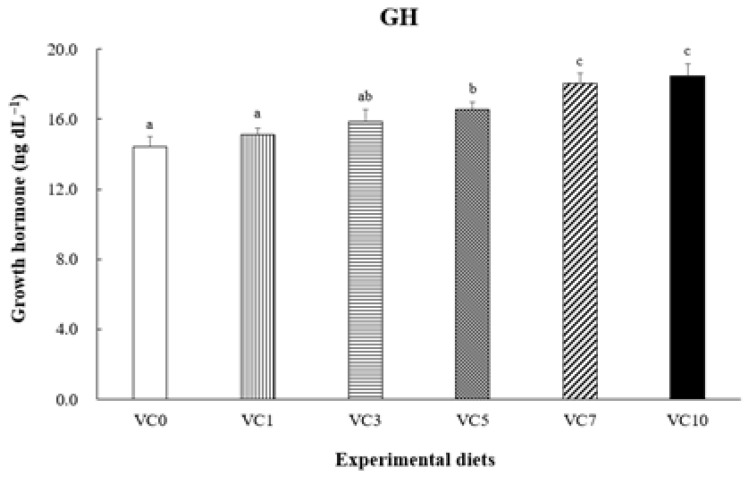
Dietary effect of supplementation of vitamin C on the growth hormone (GH) concentration in plasma of grower walleye pollock. Different letters on column indicated significant differences (*p* < 0.05).

**Figure 4 animals-14-01026-f004:**
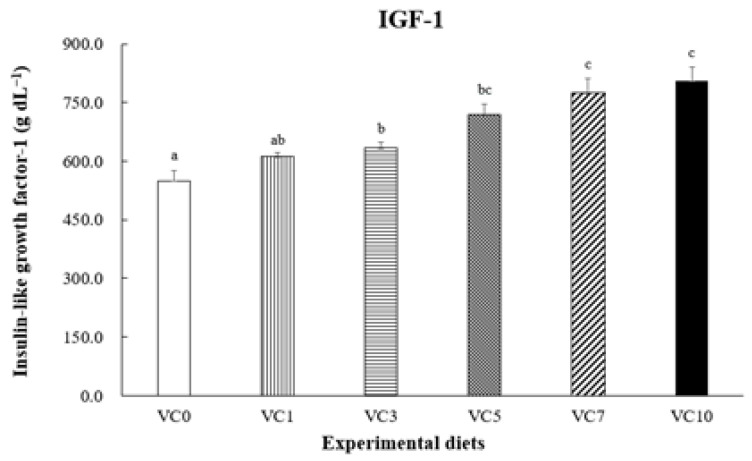
Dietary effect of supplementation of vitamin C on the insulin-like growth factor-1 (IGF-1) concentration in plasma of grower walleye pollock. Different letters on column indicated significant differences (*p* < 0.05).

**Figure 5 animals-14-01026-f005:**
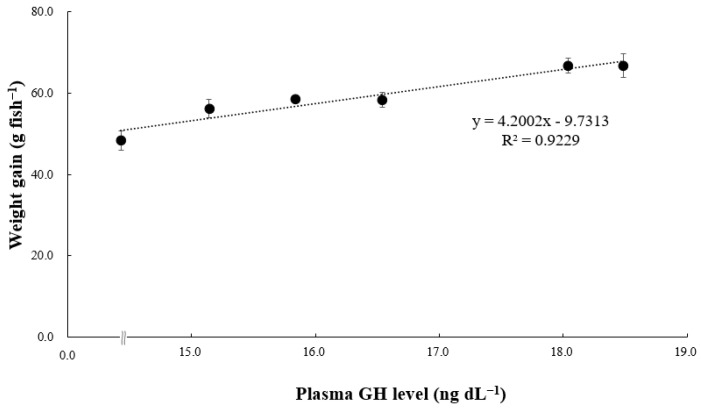
Correlation between weight gain (WG, g fish^−1^) and plasma growth hormone (GH, ng dL^−1^) level of grower walleye pollock.

**Figure 6 animals-14-01026-f006:**
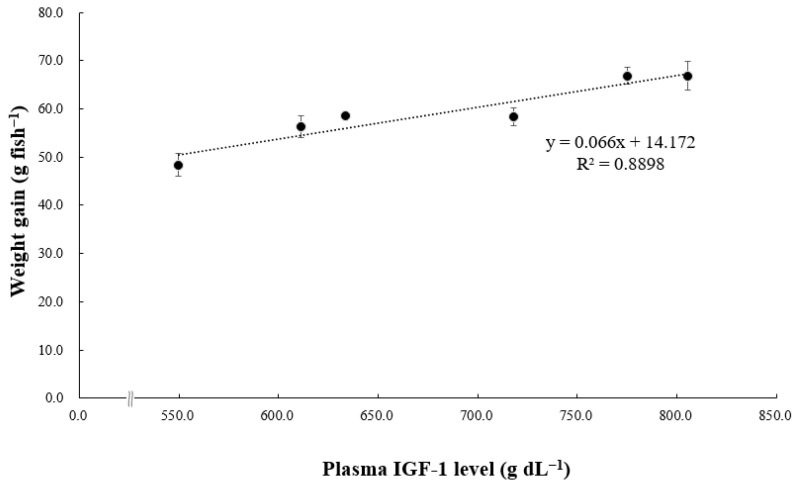
Correlation between grower walleye pollock weight gain (g fish^−1^) and insulin-like growth factor-1 (IGF-1; g dL^−1^) levels.

**Table 1 animals-14-01026-t001:** Ingredients and proximate composition of the experimental diets with different vitamin C contents (expressed as % in dry matter).

	VC0	VC1	VC3	VC5	VC7	VC10
Ingredients (%)						
Jack mackerel meal	34	34	34	34	34	34
Krill meal	40	40	40	40	40	40
Fermented soybean meal	8	8	8	8	8	8
Wheat flour	5.8	5.79	5.77	5.75	5.73	5.7
VC ^a^	-	0.01	0.03	0.05	0.07	0.1
Fish oil	8	8	8	8	8	8
Vitamin premix (VC free) ^b^	2	2	2	2	2	2
Mineral premix ^c^	2	2	2	2	2	2
Choline	0.2	0.2	0.2	0.2	0.2	0.2
Proximate composition (%)						
Moisture	94.7	94.9	96.6	96.3	95.7	96.5
Crude protein	50.7	50.2	50.5	50.2	50.1	50.4
Crude lipid	15.4	15.9	15.9	15.6	15.7	15.3
Ash	13.0	12.9	12.9	12.9	12.9	12.8
VC (mg kg^−1^)	3.24	21.92	63.31	101.42	145.46	202.51

^a^ VC: vitamin C, L-ascorbyl-2-monophosphate (Sigma-Aldrich, Germany). ^b^ Vitamin premix contained the following components (diluted in cellulose; g/kg mix): dl-α-tocopheryl acetate, 18.8; thiamin hydrochloride, 2.7; riboflavin, 9.1; pyridoxine hydrochloride, 1.8; niacin, 36.4; Ca-d-pantothenate, 12.7; myo-inositol, 181.8; d-biotin, 0.27; folic acid, 0.68; *p*-aminobenzoic acid, 18.2; menadione, 1.8; retinyl acetate, 0.73; cholecalciferol, 0.003; cyanocobalamin, 0.003. ^c^ Mineral premix contained the following ingredients (g kg^−1^ mix): MgSO_4_⋅7H_2_O, 80.0; NaH_2_PO_4_⋅2H_2_O, 370.0; KCl, 130.0; ferric citrate, 40.0; ZnSO_4_⋅7H_2_O, 20.0; Calactate, 356.5; CuCl, 0.2; AlCl_3_⋅6H_2_O, 0.15; KI, 0.15; Na_2_Se_2_O_3_, 0.01; MnSO_4_⋅H_2_O, 2.0; and CoCl_2_⋅6H_2_O, 1.0.

**Table 2 animals-14-01026-t002:** Growth performance of grower walleye pollock fed the experimental diets with different vitamin C content for 8 weeks.

Parameters	VC0	VC1	VC3	VC5	VC7	VC10	*p* Value
Initial weight (g/fish)	173.8 ± 0.36	173.1 ± 0.36	173.5 ± 0.21	173.5 ± 0.21	173.8 ± 0.36	173.1 ± 0.36	0.571
Final weight (g/fish)	222.1 ± 2.71 ^a^	229.4 ± 2.37 ^ab^	232.1 ± 0.74 ^abc^	231.3 ± 2.53 ^abc^	240.6 ± 1.65 ^c^	240.0 ± 3.22 ^bc^	0.001
Survival (%)	100.0 ± 0.00	100.0 ± 0.00	97.9 ± 2.08	100.0 ± 0.00	100.0 ± 0.00	97.9 ± 2.08	0.571
WG (g/fish)	48.3 ± 2.35 ^a^	56.3 ± 2.25 ^ab^	58.6 ± 0.54 ^bc^	58.4 ± 1.82 ^bc^	66.9 ± 1.80 ^c^	66.8 ± 2.96 ^c^	<0.001
SGR (%)	0.44 ± 0.02 ^a^	0.50 ± 0.02 ^b^	0.52 ± 0.00 ^b^	0.51 ± 0.02 ^b^	0.58 ± 0.01 ^c^	0.58 ± 0.02 ^c^	<0.001
FI (g/fish)	51.5 ± 1.68 ^a^	60.7 ± 3.09 ^ab^	62.8 ± 2.37 ^b^	60.7 ± 2.85 ^ab^	64.9 ± 0.78 ^b^	64.9 ± 2.41 ^b^	0.015
FE	0.97 ± 0.03 ^a^	0.97 ± 0.02 ^a^	1.00 ± 0.02 ^a^	1.00 ± 0.05 ^ab^	1.08 ± 0.01 ^b^	1.09 ± 0.02 ^b^	0.029
PER	1.85 ± 0.034 ^a^	1.85 ± 0.024 ^a^	1.85 ± 0.058 ^a^	1.92 ± 0.060 ^ab^	2.06 ± 0.045 ^c^	2.04 ± 0.015 ^bc^	0.008

Values are means ± SE (n = 3). Values with different superscript letters within a row are significant different (*p* < 0.05), while mean values in the same row without any superscript are not different. WG, weight gain; SGR, specific growth rate; FI, feed intake; FE, feed efficiency; PER, protein efficiency ratio.

**Table 3 animals-14-01026-t003:** Proximate composition of grower walleye pollock fed the experimental diets with different vitamin C content for 8 weeks.

Composition	VC0	VC1	VC3	VC5	VC7	VC10	*p* Value
Moisture	73.8 ± 0.27	73.6 ± 0.37	72.5 ± 0.33	73.2 ± 0.22	72.7 ± 0.50	73.1 ± 0.46	0.224
Crude protein	16.3 ± 0.12 ^a^	16.6 ± 0.09 ^ab^	16.8 ± 0.15 ^bc^	17.2 ± 0.43 ^cd^	18.0 ± 0.12 ^d^	17.8 ± 0.12 ^d^	<0.001
Crude lipid	6.5 ± 0.57	6.6 ± 0.21	6.5 ± 0.27	6.7 ± 0.24	6.6 ± 0.35	6.3 ± 0.32	0.979
Ash	2.2 ± 0.37	2.3 ± 0.15	2.2 ± 0.41	2.1 ± 0.36	2.2 ± 0.42	2.2 ± 0.31	0.492

Values are means ± SE (n = 3). Values with different superscript letters within a row are significant different (*p* < 0.05), while mean values in the same row without any superscript are not different.

**Table 4 animals-14-01026-t004:** Hematological parameters of grower walleye pollock fed the experimental diets with different vitamin C content for 8 weeks.

Parameters	VC0	VC1	VC3	VC5	VC7	VC10	*p* Value
AST (U/L)	21.0 ± 6.50	27.1 ± 7.02	39.4 ± 10.76	43.8 ± 15.53	18.6 ± 2.89	30.3 ± 8.49	0.395
ALT (U/L)	7.6 ± 1.72	6.2 ± 0.29	7.7 ± 1.02	6.6 ± 0.97	6.1 ± 0.44	7.0 ± 0.19	0.765
TCHO (mg/dL)	280.0 ± 10.97	328.1 ± 16.02	326.0 ± 19.68	325.3 ± 20.88	327.9 ± 16.15	337.9 ± 10.10	0.229
GLU (mg/dL)	81.3 ± 13.35	79.1 ± 10.27	81.7 ± 11.08	82.0 ± 6.05	93.9 ± 11.11	90.0 ± 9.18	0.098
TP (g/dL)	3.9 ± 0.13	4.2 ± 0.05	4.3 ± 0.20	4.1 ± 0.24	4.3 ± 0.17	4.4 ± 0.09	0.251

Values are means ± SE (n = 3). Values in the same row without any superscript are not different. AST, aspartate aminotransferase; ALT, alanine aminotransferase; TCHO, total cholesterol; GLU, glucose; TP, total protein.

## Data Availability

Data are available upon reasonable request.

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
