# Peer review of "Effect of Dietary Vitamin C Supplementation on Growth Performance and Biochemical Parameters in Grower Walleye Pollock, Gadus chalcogrammus"

_animals, 2024, doi:10.3390/ani14071026_

Round 1
Reviewer 1 Report
Comments and Suggestions for Authors
General comments:
This study the effect of dietary VC levels on the growth performance and biochemical parameters of grower walleye pollock and determine the optimal VC level for their diet. The results showed that the optimal dietary VC level for grower walleye pollock was approximately 156.42 mg kg−1 diet.
The manuscript could be considered for publication after being minor revisions. Some information needs to be provided as in the attachment manuscript. The format needs to be standardized as in the context of the manuscript.

Author Response
This study the effect of dietary VC levels on the growth performance and biochemical parameters of grower walleye pollock and determine the optimal VC level for their diet. The results showed that the optimal dietary VC level for grower walleye pollock was approximately 156.42 mg kg−1 diet.
The manuscript could be considered for publication after being minor revisions. Some information needs to be provided as in the attachment manuscript. The format needs to be standardized as in the context of the manuscript.
Response: We appreciate the reviewer’s comment. Thank you for the reviewer's compliment and encouragement of our study.
1.Line 41. Where is the seventh reference?
Response 1: We appreciate the reviewer’s comment. As the reviewer opinion, we revised the point.
2.Table2. Where is the note of Mineral premixC?
Response 2: Thank you for reviewer’s careful point. As the reviewer opinion, we added the note of Mineral premix c
3.Line89. Should the total fish individuals are 288? 26*3*6=468?
Response 3: We appreciate the reviewer’s comment. Checked as the reviewer’s comment. We revised the point.
4.Table 4. rearranged since some numbers are near.
Response 4: Thank you for reviewer’s careful point. We revised the point.
5.Table 4. It should be rearranged.
Response 5: Thank you for reviewer’s careful point. We revised the point.
6.Table 4. Rearranged
Response 6: Thank you for reviewer’s careful point. We revised the point.
Reviewer 2 Report
Comments and Suggestions for Authors
This study is based on a simple trial set-up, which aims to show the effects of increasing levels of dietary Vitamin C supplementation on walleye pollock performance. Vitamin C requirements have been determined in numerous species, thus this study is not especially novel; nevertheless, the following offers some suggestions, in how to improve the manuscript.
It might be slightly confusing that Vitamin C (VC) is used interchangeably between L‐ascorbyl‐2‐monophosphate on one hand and the active component ascorbic acid on the other hand. According to Material and Methods, six diets were formulated, VC0, VC1, VC3, VC5, VC7, and VC10, containing increasing VC levels: 0%, 0.01%, 0.03%, 0.05%, 0.07%, and 0.10%, respectively – but only roughly 20% were actually measured as active component. But commercial products such as Stay-C35 which is L‐ascorbyl‐2‐monophosphate contain 35% active ascorbic acid; why this discrepancy? Might this be an analytical error?
It is very surprising to see supplementation of Vit C impacted growth in such a short time, but nevertheless, fish did not even double their weight, and results might have be different if the trial would have lasted longer to replete Vit C storage. In addition, to provide more solid data, it would have been advisable to feed the fish a Vitamin C free diet ahead of the trial, and moreover to analyse the tissue analyses for changes in ascorbic acid storage.
Remarks in detail
Line 38 - reference [3] not suitable. It mentions predators such as seal and seabirds that prey on walleye
Line 40 – reference [7] missing in text, but appears in bibliography
Line 68 – reference [25] is not actually the original reference, but refers to earlier publications
Line 90 – is this correct? It says ‘In total, 288 walleye pollock were randomly stocked into 12, 400 L flow-through circular tanks (water volume: 300 L; 26 fish per tank) in triplicate’ if triplicate tanks used there should be 18 tanks ? and if 26 fish per tank are stocked, this would make a total of 468 ?
Line 101 – where is the protein retention data?
Line 132 - ‘Weight gain (WG; %) = (final body weight – initial body weight) / initial body weight’ but in table weight gain was given as g/fish not in %
Line 133 - SGR (%/day) = ‘ [(ln final weight of fish – ln initial weight of fish) / days of feeding] × 100’; it should say days of trial, not days of feeding
Line 135 ‘Feed intake (FI, g/fish) = total dry feed intake / number of surviving fish’; feed intake should be corrected for mortality, as those fish were consuming feed until mortality
Line 150 – 154 - no need to list final weight, weight gain, and SGR, as they are all descriptors of growth;
Table 2 –statistical differences or outcome should be the same for all those parameters, because as stated, they are all descriptors of growth
Line 175 ‘ Protein content was significantly different, but dietary VC did not affect moisture, crude lipid, and ash content in pollock whole-bodies’. The differing protein content might be due to analytical errors, as only the protein content changed; usually ash and protein content stay constant, and moisture content and lipid are inter- related (lower moisture = higher lipid content); also the sum of the components moisture, protein, lipid and is higher than 100%, which again might suggest an error in analytics.
Line 251 – please rephrase – this is not clear – how can a higher body protein content improve growth ?
Author Response
This study is based on a simple trial set-up, which aims to show the effects of increasing levels of dietary Vitamin C supplementation on walleye pollock performance. Vitamin C requirements have been determined in numerous species, thus this study is not especially novel; nevertheless, the following offers some suggestions, in how to improve the manuscript.
Response: We appreciate the reviewer’s comment. Thank you for the reviewer's compliment and encouragement of our study.
It might be slightly confusing that Vitamin C (VC) is used interchangeably between L‐ascorbyl‐2‐monophosphate on one hand and the active component ascorbic acid on the other hand. According to Material and Methods, six diets were formulated, VC0, VC1, VC3, VC5, VC7, and VC10, containing increasing VC levels: 0%, 0.01%, 0.03%, 0.05%, 0.07%, and 0.10%, respectively – but only roughly 20% were actually measured as active component. But commercial products such as Stay-C35 which is L‐ascorbyl‐2‐monophosphate contain 35% active ascorbic acid; why this discrepancy? Might this be an analytical error?
Response: In our opinion, the discrepancy in measuring the Vitamin C content lower than expected could arise due to various reasons such as oxidation, degradation, incomplete reaction of reagents, limitations in analytical methods, differences in manufacturing processes, or errors during measurement. Further experiments may be needed to identify the exact cause, adjusting experimental conditions and improving measurement techniques to enhance accuracy.
It is very surprising to see supplementation of Vit C impacted growth in such a short time, but nevertheless, fish did not even double their weight, and results might have be different if the trial would have lasted longer to replete Vit C storage. In addition, to provide more solid data, it would have been advisable to feed the fish a Vitamin C free diet ahead of the trial, and moreover to analyse the tissue analyses for changes in ascorbic acid storage.
Response: We completely agree with your comment. However, it should be considered that grower walleye pollock, which is the subject of our experiment, is a cold-water fish species and its growth is a little slower. We believe that future research should be conducted by formulating and feeding diet with vitamin C free and physiological analysis of the accumulated vitamin C concentration in the body.
Specific comments:
- Line 38 - reference [3] not suitable. It mentions predators such as seal and seabirds that prey on walleye
Response 1: Thank you for reviewer’s careful point. We revised the point.
2.Line 40 – reference [7] missing in text, but appears in bibliography
Response 2: Thank you for reviewer’s careful point. We revised the point.
3.Line 68 – reference [25] is not actually the original reference, but refers to earlier publications
Response 3: We appreciate the reviewer’s comment. We revised the point.
4.Line 90 – is this correct? It says ‘In total, 288 walleye pollock were randomly stocked into 12, 400 L flow-through circular tanks (water volume: 300 L; 26 fish per tank) in triplicate’ if triplicate tanks used there should be 18 tanks ? and if 26 fish per tank are stocked, this would make a total of 468 ?
Response 4: We appreciate the reviewer’s comment. We agree with reviwer’s suggestion. Accordingly, we revised the point.
5.Line 101 – where is the protein retention data?
Response 5: Thank you for reviewer’s careful point. We agree with revier’s suggestion. Accordingly, we removed it in “2.3. Sample Collection”
6.Line 132 - ‘Weight gain (WG; %) = (final body weight – initial body weight) / initial body weight’ but in table weight gain was given as g/fish not in %
Response 6: Thank you for reviewer’s careful point. We agree with revier’s suggestion. Accordingly, we revised the point.
7.Line 133 - SGR (%/day) = ‘ [(ln final weight of fish – ln initial weight of fish) / days of feeding] × 100’; it should say days of trial, not days of feeding
Response 7: Thank you for reviewer’s careful point. We revised the point.
8.Line 135 ‘Feed intake (FI, g/fish) = total dry feed intake / number of surviving fish’; feed intake should be corrected for mortality, as those fish were consuming feed until mortality
Response 8: We agree with your comment. In this study, the feed intake was corrected for mortality, and the formula for feed intake was revised as it could be confusing.
9.Line 150 – 154 - no need to list final weight, weight gain, and SGR, as they are all descriptors of growth;
Response 9: Thank you for reviewer’s careful point. However, in this study, the values of the growth parameters were statistically different from each other, so we separated them.
10.Table 2 –statistical differences or outcome should be the same for all those parameters, because as stated, they are all descriptors of growth
Response 10: Thank you for reviewer’s careful point. However, the growth parameters calculated in this study were derived from different formulas, which may result in different statistical differences. This is supported by a number of studies presented below.
Liang, X. P., Li, Y., Hou, Y. M., Qiu, H., & Zhou, Q. C. (2017). Effect of dietary vitamin C on the growth performance, antioxidant ability and innate immunity of juvenile yellow catfish (Pelteobagrus fulvidraco R ichardson). Aquaculture Research, 48(1), 149-160.
Saleh, N. E., Wassef, E. A., Kamel, M. A., El-Haroun, E. R., & El-Tahan, R. A. (2022). Beneficial effects of soybean lecithin and vitamin C combination in fingerlings gilthead seabream (Sparus aurata) diets on; fish performance, oxidation status and genes expression responses. Aquaculture, 546, 737345.
Yu, Y. B., Park, H. J., & Kang, J. C. (2020). Effects of dietary ascorbic acid on growth performance, hematological parameters, antioxidant and non-specific immune responses in starry flounder, Platichthys stellatus. Aquaculture reports, 18, 100419.
Zhao, Y., Zhao, J., Zhang, Y., & Gao, J. (2017). Effects of different dietary vitamin C supplementations on growth performance, mucus immune responses and antioxidant status of loach (Misgurnus anguillicaudatus Cantor) juveniles. Aquaculture research, 48(8), 4112-4123.
Huang, Q., Zhang, S., Du, T., Yang, Q., Chi, S., Liu, H., ... & Tan, B. (2020). Modulation of growth, immunity and antioxidant‐related gene expressions in the liver and intestine of juvenile Sillago sihama by dietary vitamin C. Aquaculture nutrition, 26(2), 338-350.
11.Line 175 ‘ Protein content was significantly different, but dietary VC did not affect moisture, crude lipid, and ash content in pollock whole-bodies’. The differing protein content might be due to analytical errors, as only the protein content changed; usually ash and protein content stay constant, and moisture content and lipid are inter- related (lower moisture = higher lipid content); also the sum of the components moisture, protein, lipid and is higher than 100%, which again might suggest an error in analytics.
Response 11: Thank you for reviewer’s careful point. After reviewing the raw data, we noticed that the moisture content was incorrectly entered. we have corrected the data in Table 1 accordingly. Also, according to previous study, the protein content in whole-body of Chu’s croaker (Nibea coibor) was showed significant differences betweent the fish fed the vitamin C treatments (Zou et al., 2020).
Zou, W., Lin, Z., Huang, Y., Limbu, S.M., Rong, H., Yu, C., … & Wen, X. (2020). Effect of dietary vitamin C on growth performance, body composition and biochemical parameters of juvenile Chu’s croaker (Nibea coibor). Aquaculture Nutrition, 26(1), 60-73.
12.Line 251 – please rephrase – this is not clear – how can a higher body protein content improve growth ?
Response 12: Thank you for reviewer’s careful point. In addition to the information provided in the text (Line 199), it is believed that the suitable inclusion of vitamin C in the feed enhances the activity of digestive enzymes and facilitates the absorption of nutrients (lipids, proteins). Several previous studies related to this phenomenon have also been documented.
→ Dietary vitamin C levels of 400 or 600 mg/kg in the feed can increase the activity of amylase, lipase, and protease in the intestine of Nile tilapia (El Basuini et al., 2021). Compared with the group without vitamin C supplementation, the activities of amylase, lipase, and protease in the intestine and liver of largemouth bass increased significantly after vitamin C supplementation (Yusuf et al., 2021). In this experiment, it was found that the amylase, lipase, and trypsin activities of liver was significantly increased by adding 100 mg/kg VC to the feed. Amylase, lipase, and trypsin can decompose starch, fat and protein into small molecules that can be absorbed.
Reviewer 3 Report
Comments and Suggestions for Authors
This study was to determine the VC requirement for grower walleye Pollock. This is a very traditional and very simple experiment. But the authors determined the requirement for VC based on fish growth alone. And, the data of this study are too simple. Although there are no obvious problems with the study, it seems to be unsuitable for publication in Animals.
L57-58 In the section of introduction, it would be useful to describe the previous VC requirements of aquatic animals, especially fish species. Although it had been discussed in the section of discussion, it should be described in briefly.
In table 1 Please provide the note of “c” for Mineral premix.
L101 Protein retention is not provided in table 2.
L105-106 Blood samples are centrifuged as serum, not plasma.
L193 Why was only SOD measured for the antioxidant index.
L201 A positive linear correlations between WG and IGF-1 level is not provided.
L257-259 Are these results consistent with previous studies on other fish species, and please explain why these hematological parameters did not change significantly in this study.
Comments on the Quality of English LanguageNo.
Author Response
This study was to determine the VC requirement for grower walleye Pollock. This is a very traditional and very simple experiment. But the authors determined the requirement for VC based on fish growth alone. And, the data of this study are too simple. Although there are no obvious problems with the study, it seems to be unsuitable for publication in Animals.
Response: We appreciate the reviewer’s comment. Thank you for the reviewer's encouragement of our study.
1.L57-58 In the section of introduction, it would be useful to describe the previous VC requirements of aquatic animals, especially fish species. Although it had been discussed in the section of discussion, it should be described in briefly.
Response 1: Thank you for reviewer’s careful point. We added the point.
2.In table 1 Please provide the note of “c” for Mineral premix.
Response 2: Thank you for reviewer’s careful point. We agree with revier’s suggestion. Accordingly, we added the note of “c” for Mineral premix.
3.L101 Protein retention is not provided in table 2.
Response 3: Thank you for reviewer’s careful point. We agree with revier’s suggestion. Accordingly, we removed it in “2.3. Sample Collection”
4.L105-106 Blood samples are centrifuged as serum, not plasma.
Response 4: We appreciate the reviewer’s comment. In this study, blood sampling from the caudal vein using a heparinized syringe, so the blood was cetrifuged to separate the plasma.
5.L193 Why was only SOD measured for the antioxidant index.
Response 5: While I agree that further analysis of the antioxidant effect should have been conducted, our research on pollock revealed a significant challenge: the slow growth rate of this cold-water fish species. To address this issue, we investigated whether vitamin C influences pollock growth. Our study primarily focused on determining the growth-improving effect of vitamin C, particularly in immature fish stages.
Given our study's primary objective, minimal attention was given to antioxidant analysis, partly due to budget constraints. However, we recognize the importance of comprehensive analysis for future research endeavors. As we explore the effects of vitamin C in other fish species, we intend to conduct thorough investigations into the antioxidant index.
In summary, while our study aimed to assess the growth-enhancing properties of vitamin C, we acknowledge the necessity for broader analysis in future research, both in terms of growth enhancement and antioxidant effects.
6.L201 A positive linear correlations between WG and IGF-1 level is not provided.
Response 6: Thank you for reviewer’s careful point. We agree with revier’s suggestion. Accordingly, we added the point.
7.L257-259 Are these results consistent with previous studies on other fish species, and please explain why these hematological parameters did not change significantly in this study.
Response 7: Thank you for reviewer’s careful point. The vitamin C content of the experimental diets in this study does not appear to affect the hematologic health of grower walleye pollock. A wider range of dietary vitamin C dosage studies are needed to determine if there is a significant effect on the blood health of grower walleye pollock. This is supported by the previous study presented below.
Ibrahim, R. E., Ahmed, S. A., Amer, S. A., Al-Gabri, N. A., Ahmed, A. I., Abdel-Warith, A. W. A., … & Metwally, A. E. (2020). Influence of vitamin C feed supplementation on the growth, antioxidant activity, immune status, tissue histomorphology, and disease resistance in Nile tilapia, Oreochromis niloticus. Aquaculture Reports, 18, 100545.
Round 2
Reviewer 3 Report
Comments and Suggestions for Authors
No.
Comments on the Quality of English LanguageNo.